# Prognostic Value of Early Intermittent Electroencephalography in Patients after Extracorporeal Cardiopulmonary Resuscitation

**DOI:** 10.3390/jcm9061745

**Published:** 2020-06-04

**Authors:** Yong Oh Kim, Ryoung-Eun Ko, Chi Ryang Chung, Jeong Hoon Yang, Taek Kyu Park, Yang Hyun Cho, Kiick Sung, Gee Young Suh, Jeong-Am Ryu

**Affiliations:** 1Department of Emergency Medicine, Dankook University Hospital, Dankook University School of Medicine, Cheonan 31116, Korea; ggggmmmmaaaail@gmail.com; 2Department of Critical Care Medicine, Samsung Medical Center, Sungkyunkwan University School of Medicine, Seoul 06351, Korea; ryoungeun.ko@samsung.com (R.-E.K.); icu.chung@samsung.com (C.R.C.); jhysmc@gmail.com (J.H.Y.); suhgy@skku.edu (G.Y.S.); 3Division of Cardiology, Department of Medicine, Samsung Medical Center, Sungkyunkwan University School of Medicine, Seoul 06351, Korea; taekkyu.park@samsung.com; 4Department of Thoracic and Cardiovascular Surgery, Samsung Medical Center, Sungkyunkwan University School of Medicine, Seoul 06351, Korea; yanghyun.cho@samsung.com (Y.H.C.); kiick.sung@samsung.com (K.S.); 5Division of Pulmonary and Critical Care Medicine, Department of Medicine, Samsung Medical Center, Sungkyunkwan University School of Medicine, Seoul 06351, Korea; 6Department of Neurosurgery, Samsung Medical Center, Sungkyunkwan University School of Medicine, Seoul 06351, Korea

**Keywords:** cardiopulmonary resuscitation, extracorporeal membrane oxygenation, neurological prognosis, electroencephalography

## Abstract

The aim of this study was to investigate whether early intermittent electroencephalography (EEG) could be used to predict neurological prognosis of patients who underwent extracorporeal cardiopulmonary resuscitation (ECPR). This was a retrospective and observational study of adult patients who were evaluated by EEG scan within 96 h after ECPR. The primary endpoint was neurological status upon discharge from the hospital assessed with a Cerebral Performance Categories (CPC) scale. Among 69 adult cardiac arrest patients who underwent ECPR, 17 (24.6%) patients had favorable neurological outcomes (CPC score of 1 or 2). Malignant EEG patterns were more common in patients with poor neurological outcomes (CPC score of 3, 4 or 5) than in patients with favorable neurological outcomes (73.1% vs. 5.9%, *p* < 0.001). All patients with highly malignant EEG patterns (43.5%) had poor neurological outcomes. In multivariable analysis, malignant EEG patterns and duration of cardiopulmonary resuscitation were significantly associated with poor neurological outcomes. In this study, malignant EEG patterns within 96 h after cardiac arrest were significantly associated with poor neurological outcomes. Therefore, an early intermittent EEG scan could be helpful for predicting neurological prognosis of post-cardiac arrest patients after ECPR.

## 1. Introduction

Neurological prognosis is one of the most important issues in patients who survive a cardiac arrest [1,2]. It is important to estimate the reversibility of cerebral function in patients after return of spontaneous circulation. It may prevent inappropriate continuation of intensive treatment in patients who are predicted to have poor neurological outcomes [2,3]. Recently, extracorporeal membrane oxygenation (ECMO) has been increasingly used as an adjuvant therapy of conventional cardiopulmonary resuscitation (CPR), providing oxygenated blood and hemodynamic support in the absence of spontaneous cardiac circulation [4,5].

Autoregulation of cerebral blood flow may be changed in patients resuscitated from cardiac arrest [6]. It is difficult to predict how highly oxygenated continuous flow by ECMO affects cerebral autoregulation and neurological recovery in the setting of extracorporeal cardiopulmonary resuscitation (ECPR) [2]. In previous studies of ECPR, several predictors of mortality have been reported. However, limited data are available on neurological prognosis after ECPR [7].

Among electrophysiologic studies, electroencephalography (EEG) has been most widely used as one of the assessment tools for survivors after cardiac arrest [8]. In the setting of ECPR, whether an early intermittent EEG scan may be helpful for systemically estimating neurological outcomes of survivors has not been reported yet. Therefore, the objective of this study was to investigate whether an early intermittent EEG scan could be used to predict neurological outcomes of patients who underwent ECPR.

## 2. Methods

### 2.1. Study Population and Design

This was a retrospective, single-center, observational study of adult patients who underwent ECPR during hospitalization between February 2012 and December 2018. This study was approved by the Institutional Review Board of Samsung Medical Center (IRB no. SMC 2019-05-002). The requirement for informed consent was waived due to its retrospective nature. Clinical and laboratory data were collected by a trained study coordinator using a standardized case report form. Inclusion criteria were: (1) those who underwent ECPR during the study period; (2) those who had decreased mentalities (a score of <13 on the Glasgow Coma Scale) on EEG scan after cardiac arrest; and (3) those whose EEG scans were performed within 96 h after ECPR. Exclusion criteria were: (1) those who were under 18 years of age; (2) those with malignancy whose life expectancy was less than 1 year; (3) those with insufficient medical records; (4) those with causes of death verified to be other than brain death; and (5) those with a history of head trauma or a chronic neurological abnormality upon admission to the intensive care unit (ICU). Ultimately, a total of 69 patients with an EEG scan who were resuscitated by veno-arterial ECMO were analyzed in this study (Figure 1).

### 2.2. Endpoints and Definitions

In this study, ECPR was defined as a successful veno-arterial ECMO implantation and pump-on with chest compression for external cardiac massage during index procedure in patients with cardiac arrest. When a return of spontaneous circulation occurs during ECMO cannulation, practitioners typically do not remove the cannula or stop the ECMO pump-on process [7]. Surface cooling and the degree of targeted temperature were determined by each intensivist in the ICU according to the targeted temperature management protocol [9]. The primary endpoint was the neurological status on discharge from the hospital. It was assessed with the Glasgow–Pittsburgh Cerebral Performance Categories (CPC) scale (scores range from 1 to 5) [10]. CPC scores of 1 and 2 were classified as favorable neurological outcomes while CPC scores of 3, 4 and 5 were considered as poor neurological outcomes [11,12]. Medical records were thoroughly reviewed. Patients were graded on the CPC score by two independent neurologists. If the CPC scores did not match between the two neurologists, they discussed it and reached an agreement. A neurointensivist, attending physician or consultant neurologist determined EEG scan. An EEG scan was performed to identify causes of decreased consciousness or to predict neurological outcomes in ECPR patients. However, an EEG scan was not performed for patients who had a rapid recovery of mentality and neurological deficits. An EEG scan was also performed when patients had accompanied seizures or abnormal movements. If sedatives, analgesics or antiepileptic drugs were administrated to patients after ECPR, these drugs were not stopped during the EEG scan. EEG was performed using a 64-channel digital video-EEG system (Nicolet Biomedical, Inc., Madison, WI, USA). Surface electrodes were placed according to the international 10–20 system. Additional electrodes were placed whenever needed [13]. EEG patterns of ECPR patients were defined using the EEG terminology of the American Clinical Neurophysiology Society [14,15]. Malignant EEG patterns were defined as highly malignant EEG patterns and moderate malignant EEG patterns. Highly malignant EEG patterns were defined as suppressed background (amplitude <10 μV, 100% of the recording) without discharges, suppressed background with superimposed continuous periodic discharges, or burst-suppression (periods of suppression with amplitude <10 μV constituting >50% of the recording) with superimposed discharges or without discharges [15] (Figure 2).

Moderate malignant EEG patterns were defined as malignant periodic or rhythmic patterns (abundant periodic discharges; abundant rhythmic spikes, polyspikes, sharp waves, spike-and-wave or sharp-and-slow wave; unequivocal electrographic seizure), malignant background (discontinuous background; low voltage background; reversed anterior-posterior gradient) or unreactive EEG (absence of background reactivity or only stimulus-induced discharges) [15]. Benign EEG patterns were defined as an absence of all malignant features stated above. EEG findings were confirmed by three EEG specialists.

### 2.3. Procedure

CPR was led by the CPR team of the hospital. All facts related to the CPR scene were recorded by bedside nurses according to Utstein-Style guidelines [16]. When CPR was performed for more than 10 min or in the event of unstable vital signs or recurrent cardiac arrest, the institutional rapid response team contacted the on-call ECMO team leader, who along with the CPR leader assessed the patient and made a decision about whether to institute ECPR. ECPR was performed when a witnessed arrest was confirmed, when the arrest persisted despite conventional CPR lasting for more than 10 min, and when the event that caused the arrest was considered reversible [4]. Cases in which ECPR was deferred included those with a short life expectancy (<6 months), terminal malignancy, an unwitnessed collapse, limited physical activity or CPR undertaken for more than 60 min at the time of initial contact. Age alone did not constitute a contraindication to ECPR [4].

The ECMO team consisted of cardiologists, cardiovascular surgeons, intensivists, special nurses and perfusionists. Either a Capiox Emergency Bypass System (Terumo, Tokyo, Japan) or a Prolonged Life Support System (Maquet Cardiopulmonary, Hirrlingen, Germany) was used in all cases. A crystalloid solution such as normal saline or balanced solution was used for priming. No patient had blood-primed ECMO. A percutaneous vascular approach was tried initially in all cases using the Seldinger technique. When percutaneous cannulation failed, surgical cutdown exposure was performed [4]. Femoral vessels were the most common sites of vascular access using 14 to 17 French arterial cannulas and 20 to 24 French venous cannulas [7]. Cardiac compression was stopped once ECMO pump-on was successful during CPR. Anticoagulation was accomplished by a bolus injection of unfractionated heparin, followed by continuous intravenous heparin infusion to maintain an activated clotting time between 150 and 180 s. The initial number of revolutions per minute of the ECMO device was adjusted to achieve an ideal cardiac index greater than 2.2 L/min/m^2^ of body surface area, central mixed venous oxygen saturation above 70%, and a mean arterial pressure above 65 mm Hg [7]. Blood pressure was monitored continuously through an arterial catheter. An artery in the right arm was used for arterial blood gas analysis to estimate cerebral oxygenation. After ECMO, necessary steps were taken to treat the cause of the arrest, such as percutaneous coronary intervention, coronary artery bypass grafting, heart transplantation, non-coronary cardiopulmonary surgery or non-cardiopulmonary surgery [7].

### 2.4. Statistical Analyses

All data are presented as medians and interquartile ranges (IQRs, Q1~Q3) for continuous variables and as numbers (percentages) for categorical variables. Data were compared using the Mann–Whitney *U* test for continuous variables and the Chi-squared test or Fisher’s exact test for categorical variables. Variables with *p* values less than 0.2 in univariate analyses and clinically relevant variables were subjected to a stepwise multiple logistic regression model to obtain statistically meaningful predictor variables. They were EEG groupings by its pattern, age, target temperature management, first monitored rhythm, CPR duration, Glasgow Coma Scale on EEG scan and use of sedative or analgesic. Due to small event rates, we took the caution of the general rule of 10 events per variable before any routine application of statistical methods and applied the Firth’s correction. Adequacy of the prediction model was determined using the Hosmer–Lemeshow test, along with the areas under the curve (AUCs). The predictive performance of malignant EEG patterns assessed using the AUCs of the receiver operating characteristic (ROC) curves for sensitivity vs. 1-specificity. The AUCs were compared using the nonparametric approach published by DeLong et al. [17] for two correlated AUCs. All tests were two-sided and *p* < 0.05 was considered statistically significant. All data were analyzed using IBM SPSS version 20 (IBM, Armonk, NY, USA) and R Statistical Software (version 3.6.3; R Foundation for Statistical Computing, Vienna, Austria).

## 3. Results

### 3.1. Baseline Characteristics and Clinical Outcomes

The median patient age was 56 (IQR: 47–70) years. Of 69 patients included in this study, 52 (75.4%) were males. Hypertension (42.0%) and diabetes mellitus (33.3%) were the most common comorbidities among patients who underwent ECPR. Hypertension was more common in patients with poor neurological outcomes than in patients with favorable neurological outcomes (50.0% vs. 17.6%, *p* = 0.005). A cardiac cause of arrest was verified in 59 (85.5%) patients. Acute coronary syndrome was the main cause of cardiac arrest in 26 (44.1%) patients. Fourteen (20.3%) patients had a history of ischemic heart disease. Forty-seven (68.1%) patients experienced cardiac arrest in the hospital while 22 (31.9%) patients suffered cardiac arrest in an out-of-hospital setting. Compared with the group with favorable neurological outcomes, the group with poor neurological outcomes had a longer CPR duration (*p* = 0.005). Baseline characteristics of ECPR patients are presented in Table 1.

Among the 69 adult cardiac arrest patients who underwent ECPR, 32 (46.4%) survived until discharge from the hospital. Of these 32 survivors, 17 (24.6%) had favorable neurological outcomes (CPC score of 1 or 2). The entire distribution of CPC scores is shown in Figure 1.

### 3.2. Relationship between EEG and Neurologic Outcomes

Sedatives or analgesics were used in 41 (59.4%) patients who underwent ECPR. These drugs were used more in patients with favorable neurological outcomes than in patients with poor neurological outcomes (88.2% vs. 50.0%, *p* = 0.012). There was no significant difference in the use of antiepileptic drugs between the two groups of patients (*p* = 0.999). Characteristics on the EEG scan are presented in Table 2.

Malignant EEG patterns were more common in patients with poor neurological outcomes than in patients with favorable neurological outcomes (73.1% vs. 5.9%, *p* < 0.001, Table 3). All patients with highly malignant EEG patterns (43.5%) had poor neurological outcomes. Moderately malignant EEG patterns were reported in eight (11.6%) patients with poor neurological outcomes and in only one (1.4%) patient with a favorable neurological outcome. Regardless of the interval between ECPR and EEG scans, most patients with malignant EEG patterns had poor neurological outcomes in this study. In addition, all patients with myoclonic status epilepticus had poor neurological outcomes. Benign EEG patterns were more common in patients with favorable neurological outcomes than in patients with poor neurological outcomes (94.1% vs. 26.9%, *p* < 0.001, Table 3).

In multivariable analysis, the only significant indicators were EEG grouping by its pattern and CPR duration. That is, malignant EEG patterns (adjusted odd ratio (OR): 36.43, 95% confidence interval (CI): 4.632–1013.711) and CPR duration (adjusted OR: 1.06 per minute increase, 95% CI: 1.00–1.138) were significantly associated with poor neurological outcomes in patients who underwent ECPR (Hosmer–Lemeshow Chi-squared = 2.32, *df* = 8, *p* = 0.969) with the AUCs of 0.946 (95% CI 0.893–0.999) (Table 4).

Although there were no differences between the AUCs of malignant EEG patterns and CPR duration, the performance of a composite of these marker was strongly associated with poor neurological outcomes compared with the use of either marker alone (*p* = 0.008 and *p* = 0.006, respectively) (Figure 3).

## 4. Discussion

In this study, we investigated whether intermittent EEG could be used to predict neurological outcomes of patients who underwent ECPR. The major findings of this study were as follows. First, regardless of sedation, malignant EEG patterns were more common in patients with poor neurological outcomes than in patients with favorable neurological outcomes. Especially, all patients with highly malignant EEG patterns had poor neurological outcomes. In addition, patients with moderate malignant EEG patterns had poor neurological outcome except for one patient. Second, benign EEG patterns alone did not necessarily imply a favorable neurological outcome. Third, in multivariable analysis, malignant EEG patterns and CPR duration were significantly associated with poor neurological outcomes in patients who underwent ECPR. Therefore, early intermittent EEG scans and CPR duration could be helpful for predicting neurological outcomes of post-cardiac arrest patients after ECPR.

EEG signals mainly reflect cerebral cortical function and some subcortical function [8]. EEG is very sensitive to ischemia because cortical neurons of the brain need a consistent blood supply to maintain signaling and integrity [8]. Therefore, the EEG scan is a standard and useful tool to predict neurological outcomes after cardiac arrest [3,15]. Especially, malignant EEG patterns such as suppressed background, status epilepticus, burst suppression, periodic patterns and unreactive EEG are associated with poor neurological prognosis after cardiac arrest [8,15,18]. In addition, early continuous waves with normal voltage could be a predictor of favorable neurological outcomes after cardiac arrest [19].

Cerebral autoregulation may be changed in survivors after cardiac arrest [6]. Highly oxygenated continuous ECMO flow could affect cerebral autoregulation after ECPR [2]. In addition, neurological outcomes may be affected by functional recovery of native heart and lung, the amount of ECMO support and changed cerebral autoregulation [2]. Altered cerebral hemodynamics by ECMO support may influence neurological outcomes after ECPR. Therefore, it is difficult to predict neurological prognosis by these changed situations after ECPR [2]. Ultimately, the interaction between cerebral autoregulation and ECMO flow may affect neurological recovery and prognosis in ECPR patients through mechanisms of primary ischemic damage and secondary additive injury [2]. Thus, EEG change by this interaction should be studied for neurological prediction after ECPR. However, there has been no report of EEG according to neurological outcomes after ECPR.

Sedation may confuse outcome predictions in survivors of cardiac arrests [1,8,20]. Sedatives are commonly used in survivors after cardiac arrest for 72 h as important confounders [1,20]. A motor response to noxious stimuli, corneal reflex, caloric testing and some electrophysiologic studies may also be confounded by sedation [20,21]. Although mild to moderate hypothermia does not significantly affect EEG in patients with induced hypothermia [8,22], a confounder accompanied by induced hypothermia such as analgesics, sedatives or artifacts from shivering, mechanical ventilator or electrical devices may affect the reliability of EEG interpretations [8]. However, a recent study has reported that the predictive performance of EEG after cardiac arrest is similar between patients with ongoing sedation and those without ongoing sedation [8,15]. In this study, sedation or targeted temperature management did not significantly affect the prediction of poor neurological outcomes after ECPR. Regardless of sedation or targeted temperature management, patients with malignant EEG patterns had poor neurological outcomes in this study.

Benign EEG patterns may be associated with a favorable neurological outcome in survivors after cardiac arrest [19]. Especially, early continuous waves with normal voltage could be a predictor of favorable neurological outcomes after cardiac arrest [8,15,19]. However, in previous studies, benign EEG patterns are not always associated with a good neurological outcome [15,18]. Additive secondary injury is characterized by an imbalance in post-resuscitation cerebral oxygen delivery and use [23]. This injury is associated with reperfusion injury, impaired autoregulation, fluctuations in oxygen support and arterial carbon dioxide, hyperthermia and concomitant anemia [23]. Early EEG findings may not be shown to be malignant EEG patterns in patients with poor neurological outcomes if the secondary cerebral injury is more serious than the primary cerebral injury. In this study, benign EEG patterns were not always associated with a favorable neurological outcome. In addition, intermittent EEG scans may be less sensitive for predicting favorable neurological outcomes than continuous EEG monitoring in this study.

It is not easy to predict neurological prognosis of survivors after cardiac arrest [24,25]. In addition, there is no optimal timing to assess prognosis after cardiac arrest [25]. Although highly malignant EEG patterns may be associated with poor prognosis, other EEG patterns may be ambiguous in predicting outcomes in various conditions after cardiac arrest [15,26]. Therefore, it may be unreasonable to predict the prognosis of survivors after ECPR based on EEG patterns alone. Early EEG scans may help to predict neurological outcomes and reduce uncertainty over coma prognostication after cardiac arrest [26]. Therefore, EEG should be used as one of the multimodal tools rather than an absolute tool for predicting prognosis. Eventually, a multimodal approach, including neurological examination, biomarker, brain imaging, evoked potential and EEG may be needed to predict neurological outcomes in patients after ECPR [24,25].

This study has several limitations. First, this was a retrospective review. Thus, CPC score was determined based on medical records. By using two independent specialists’ agreement on the score, any bias may be mitigated to some extent. In addition, although the cause of death had to be accurately verified, its identification was insufficient due to the retrospective nature of this study. Second, the nonrandomized nature of registry data might have resulted in selection bias. EEG scans were not protocol-based in their performance. Particularly, during the study period, EEG scans were not performed in all patients. They were only performed in patients with abnormal consciousness, seizure, abnormal movements or other symptoms. Although EEG scans were performed within 96 h following ECPR, a major limitation of this study might be that EEG scans were performed in different time settings. Lastly, our study has limited statistical power due to its small sample size. To alleviate the small sample issue, we applied the Firth’s correction, which dampens the results somewhat. Although it still provides a valuable insight, prospective large-scale studies are needed to confirm the usefulness of early intermittent EEG scan in predicting neurological outcomes of patients after ECPR to arrive at evidence-based conclusions.

## 5. Conclusions

In this study, malignant EEG patterns within 96 h after cardiac arrest were significantly associated with poor neurological outcomes in patients who underwent ECPR. A multimodal approach is needed to predict neurological outcomes in patients after ECPR. Especially, an early EEG scan may help to predict neurological outcomes and reduce uncertainty about coma prognostication after ECPR.

## Figures and Tables

**Figure 1 jcm-09-01745-f001:**
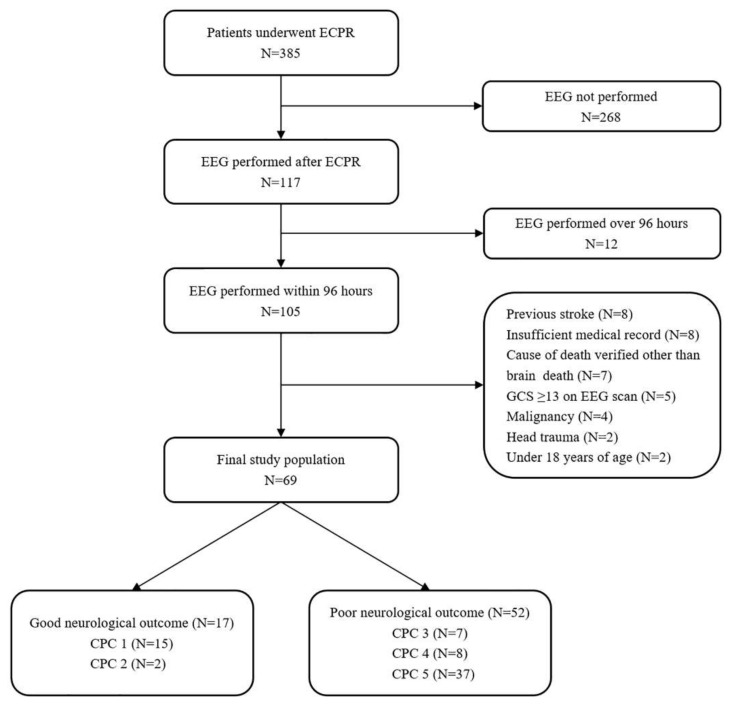
Study flow chart. ECPR, extracorporeal cardiopulmonary resuscitation; EEG, electroencephalography; GCS, Glasgow Coma Scale; CPC, Cerebral Performance Categories.

**Figure 2 jcm-09-01745-f002:**
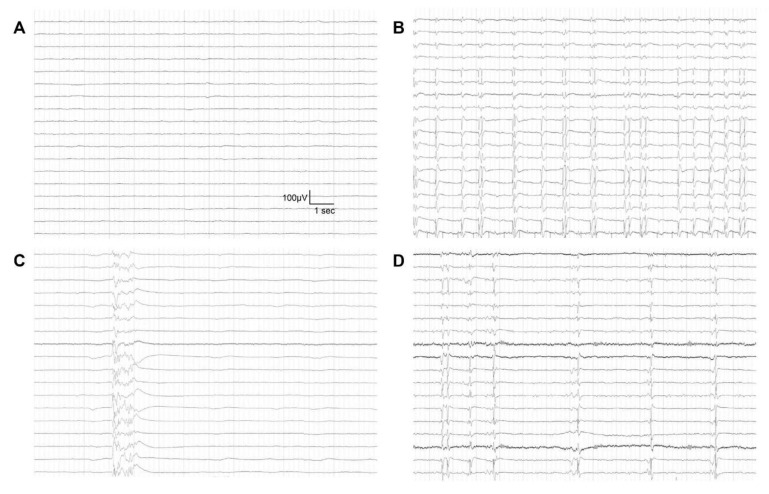
Highly malignant patterns of electroencephalography in patients after extracorporeal cardiopulmonary resuscitation. (**A**) Suppressed background without discharges; (**B**) Suppressed background with superimposed continuous periodic discharges; (**C**) burst-suppression without discharges; and (**D**) burst-suppression with superimposed discharges.

**Figure 3 jcm-09-01745-f003:**
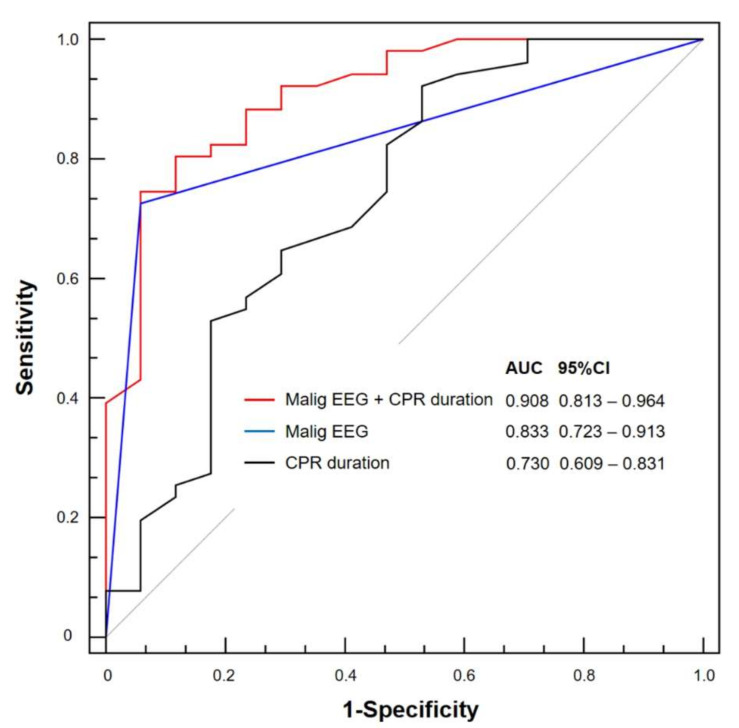
Receiver operating characteristic curves for predicting poor outcomes using malignant patterns of electroencephalography (EEG) and cardiopulmonary resuscitation (CPR) duration. Although there were no differences between the areas under the curve (AUCs) of malignant EEG patterns and CPR duration, the performance of a composite of these marker was strongly associated with poor neurological outcomes compared with the use of either marker alone (*p* = 0.008 and *p* = 0.006, respectively). CI, confidence interval; Malig EEG, malignant EEG patterns.

**Table 1 jcm-09-01745-t001:** Baseline characteristics.

	Favorable Neurological Outcome (n = 17)	Poor Neurological Outcome (n = 52)	*p* Value
Age (y)—median (IQR)	51.0 (36.0–73.0)	57.5 (50.0–69.5)	0.460
Gender, male—no. of patients (%)	15 (88.2)	37 (71.2)	0.206
Body mass index (kg/m^2^)—median (IQR)	23.5 (21.3–27.7)	25.2 (22.2–28.7)	0.354
Medical history—no. of patients (%)			
Hypertension	3 (17.6)	26 (50.0)	0.039
Diabetes mellitus	3 (17.6)	20 (38.5)	0.199
Current smoker	6 (35.3)	11 (21.2)	0.331
Previous myocardial infarction	2 (11.8)	12 (23.1)	0.491
Malignancy	1 (5.9)	11 (21.2)	0.269
Dyslipidemia	2 (11.8)	10 (19.2)	0.716
Target temperature management—no. of patients (%)	9 (52.9)	21 (40.4)	0.532
Type of cardiac arrest—no. of patients (%)			0.581
Out of hospital cardiac arrest	4 (23.5)	18 (34.6)	
In-hospital cardiac arrest	13 (76.5)	34 (65.4)	
Bystander witnessed cardiac arrest—no. of patients (%)	17 (100)	50 (96.2)	0.999
Bystander performed CPR—no. of patients (%)	16 (94.1)	48 (92.3)	0.999
First monitored rhythm—no. of patients (%)			0.167
Asystole	1 (5.9)	11 (21.2)	
Pulseless electrical activity	6 (35.3)	23 (44.2)	
Shockable rhythm (VT or VF)	10 (58.8)	18 (34.6)	
Defibrillation—no. of patients (%)	12 (70.6)	30 (57.7)	0.510
CPR duration (min)—median (IQR)	19.0 (8.0–28.0)	31.0 (21.0–40.5)	0.005
Location of ECMO insertion—no. of patients (%)			0.177
Emergency room	9 (52.9)	24 (46.2)	
Intensive care unit	4 (23.5)	23 (44.2)	
Cath room	4 (23.5)	4 (7.7)	
Operation room	0 (0)	1 (1.9)	
Cardiac cause of arrest—no. of patients (%)			0.999
Ischemic	7 (41.2)	19 (45.2)	
Non-ischemic	10 (58.8)	23 (54.8)	

IQR: interquartile range; CPR: cardiopulmonary resuscitation; VT: ventricular tachycardia; VF: ventricular fibrillation; ECMO: extracorporeal membrane oxygenation.

**Table 2 jcm-09-01745-t002:** Characteristics on electroencephalography scan.

	Favorable Neurological Outcome (n = 17)	Poor Neurological Outcome (n = 52)	*p* Value
Interval between ECPR and EEG scan—no. of patients (%)			0.671
0–12 h	2 (11.8)	10 (19.2)	
12–24 h	4 (23.5)	10 (19.2)	
24–48 h	5 (29.4)	20 (38.5)	
48–96 h	6 (35.3)	12 (23.1)	
Reasons of EEG scan no. of patients (%)			0.724
For neurological outcome prediction or decreased mentality	7 (41.2)	26 (50.0)	
Seizure or abnormal movement	10 (58.8)	26 (50.0)	
Pupil reflex—no. of patients (%)			0.199
Both prompt	13 (76.5)	27 (51.9)	
One or both sluggish	2 (11.8)	8 (15.4)	
One or both fix	2 (11.8)	17 (32.7)	
Glasgow Coma Scale on EEG scan	7.0 (3.0–9.0)	3.0 (3.0–7.0)	0.012
Use of sedative or analgesic—no. of patients (%)	15 (88.2)	26 (50.0)	0.012
Bolus infusion	4 (23.5)	10 (19.2)	0.734
Continuous infusion	15 (88.2)	24 (46.2)	0.006
Remifentanil	8 (47.1)	13 (25.0)	
Midazolam	7 (41.2)	9 (17.3)	
Fentanyl	6 (35.3)	5 (9.6)	
Propofol	4 (23.5)	2 (3.8)	
Use of antiepileptic drug	4 (23.5)	13 (25.0)	0.999

ECPR: extracorporeal cardiopulmonary resuscitation; EEG: electroencephalography.

**Table 3 jcm-09-01745-t003:** Findings of electroencephalography.

	Favorable Neurological Outcome (n = 17)	Poor Neurological Outcome (n = 52)	*p* Value
EEG findings—no. of patients (%)			<0.001
Benign EEG	16 (94.1)	14 (26.9)	
Malignant EEG	1 (5.9)	38 (73.1)	
Highly malignant EEG			
Suppressed background without discharges	0 (0)	18 (34.6)	
Suppressed background with continuous periodic discharges	0 (0)	2 (3.8)	
Burst-suppression background with or without discharges	0 (0)	10 (19.2)	
Moderately malignant EEG			
Malignant periodic or rhythmic patterns	0 (0)	6 (11.5)	
Malignant background	0 (0)	2 (3.8)	
Unreactive EEG	1 (5.9)	0 (0)	
EEG patterns according to time interval—no. of patients (%)			
EEG performed within 24 h after ECPR			0.001
Benign EEG patterns	6 (35.3)	4 (7.7)	
Malignant EEG patterns	0 (0)	16 (30.8)	
EEG performed over 24 h after ECPR			0.001
Benign EEG patterns	10 (58.8)	10 (19.2)	
Malignant EEG patterns	1 (5.9)	22 (42.3)	
Accompanied clinical seizure—no. of patients (%)			0.121
Absence of clinical seizure	7 (41.2)	28 (53.8)	
Sporadic seizure or myoclonus	10 (58.8)	18 (34.6)	
Myoclonic status epilepticus	0 (0)	6 (11.5)	

ECPR: extracorporeal cardiopulmonary resuscitation; EEG: electroencephalography.

**Table 4 jcm-09-01745-t004:** Multivariable logistic regression of clinically relevant variables associated with poor neurological outcomes.

	Adjusted OR (95% CI)	*p* Value
Malignant EEG patterns	36.43 (4.632–1013.711)	<0.001
First monitored rhythm		
Asystole	1	Reference
Pulseless electrical activity	0.84 (0.046–11.674)	0.894
Shockable rhythm (VT or VF)	0.19 (0.011–1.622)	0.135
CPR duration (min)	1.06 (1.009–1.138)	0.020
Use of sedative or analgesic	0.26 (0.023–1.845)	0.188

OR: odd ratio; CI: confidence interval; EEG: electroencephalography; VT: ventricular tachycardia; VF: ventricular fibrillation; CPR: cardiopulmonary resuscitation. Variables with *p* values less than 0.2 in univariate analyses and clinically relevant variables were subjected to a stepwise multiple logistic regression model to obtain statistically meaningful predictor variables. They were EEG groupings by its pattern, age, target temperature management, first monitored rhythm, CPR duration, Glasgow Coma Scale on EEG scan, and use of sedative or analgesic. To alleviate the small sample issue, we applied the Firth’s correction.

## Data Availability

Regarding data availability, our data are available on the Harvard Dataverse Network (http://dx.doi.org/10.7910/DVN/KYJNVA).

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
