# Peer review of "Prognostic Value of Early Intermittent Electroencephalography in Patients after Extracorporeal Cardiopulmonary Resuscitation"

_jcm, 2020, doi:10.3390/jcm9061745_

Round 1

Reviewer 1 Report

This is an interesting and important study. However, there are aspects that must be clarified. The authors state that logistic regression found CPR duration and EEG abnormalities were the only significant predictors of neurological outcome. They need to tell us all the other predictors that were tested in the model. The use of sedatives and analgesics is of particular interest. Obviously, if a patient if deeply in coma and does not require any sedation or analgesia they are likely to have more brain damage than a patient who needs to be sedated to prevent him or her waking up. The challenge for the authors is to prove that EEG prognostication trumps simply using the need for sedation and analgesia.  If possible, this should be explored in detail: for example, if the patient is experiencing pain, I would suspect that is a “good sign”. Can the authors address this? If not, then explain why.

Was the need to use sedatives and analgesics just as good in predicting outcome as EEG? Was this tested in the logistic model? If not, why not? What were the variables tested in the logistic regression model? Were they all the variables tested in the logistic regression model those in Table 1, 2 and 3? How were they handled – as categorical or continuous variables? How were EEG abnormalities handled?

There needs to be an extra table that shows exactly what variables were tested by logistic regression and what the results were (i.e. adjusted odds ratio for each variable and its p value).

The other major issue is the difficulty interpreting the Tables. I think this is an unfortunate formatting issue that might have occurred while the paper was being put into a pdf format. The numbers and the variables do not line up. This makes Table 1 and 2 impossible to interpret, and Table 3 difficult.

Minor issues:

Why excludes malignancy with life expectancy less than 1 year? I can understand why a patient might not get ECMO, but if they got it then why not tell us what their outcome was?

What happened to the 36 patients who had EEG but were excluded? Even a rough narrative would be interesting.

Fig 1 flow diagram should show how many patients made a rapid neurological recovery and did not need EEG. Was that all the 268 patients who did not get EEG?

Fig 1: define CPC in legend

In Table 1 how long was CPR duration – minutes?

Odds ratio of CPR duration is 1.07 – does that mean per minute? Please clarify.

Please use either c-statistic or AUC – its confusing if you use one and then the other.

Author Response

Thank you for your kind comments. Please see the attachment.

Reviewer 2 Report

Thank you for the opportunity to review this manuscript, which reports on a study to show that early intermittent electroencephalography (EEG) scan within 96 hours after the onset of cardiac arrest could be helpful for predicting neurological prognosis of post-cardiac arrest patients after extracorporeal cardiopulmonary resuscitation. This retrospective observational study analysed the date from a single hospital in Korea between February 2012 and December 2018. I have great concern about the data the authors analysed, therefore several issues must be addressed by the authors. The major issue of this manuscript is about the analyses of malignant EEG. Although the authors stated the inconsistent time of EEG scans in the limitation section, there were wide 95% confidence intervals (Cis) in the adjusted odds ratio of malignant EEG for predicting poor neurological outcomes: adjusted odds ratio, 53.26; 95% CI, 5.956-476.249. This may be due to not only various EEG scan time but also small number of study subjects. Therefore, more study number of cases are required with prospective settings. Any single predictor may not practical for predicting neurological outcomes: Sandroni C et al. Resuscitation 2014;85:1779-89.

Author Response

(The authors gave the same response as above.)

Round 2

Reviewer 1 Report

Thank you for your revisions.

Reviewer 2 Report

The manuscript has grately improved. I have no suggestions. Thank you for your cooperation.